# Volatile Organic Compounds in Cellular Headspace after Hyperbaric Oxygen Exposure: An In Vitro Pilot Study

**DOI:** 10.3390/metabo14050281

**Published:** 2024-05-13

**Authors:** Feiko J. M. de Jong, Thijs A. Lilien, Dominic W. Fenn, Thijs T. Wingelaar, Pieter-Jan A. M. van Ooij, Anke H. Maitland-van der Zee, Markus W. Hollmann, Rob A. van Hulst, Paul Brinkman

**Affiliations:** 1Royal Netherlands Navy Diving and Submarine Medical Centre, 1780 CA Den Helder, The Netherlands; 2Department of Anesthesiology, Amsterdam UMC, Location AMC, 1100 DD Amsterdam, The Netherlands; 3Department of Pediatric Intensive Care, Amsterdam UMC, Location Emma Children’s Hospital, 1100 DD Amsterdam, The Netherlands; 4Department of Pulmonology, Amsterdam UMC, Location AMC, 1100 DD Amsterdam, The Netherlands

**Keywords:** Treatment Table 6, GC-MS, oxidative stress, pulmonary oxygen toxicity

## Abstract

Volatile organic compounds (VOCs) might be associated with pulmonary oxygen toxicity (POT). This pilot study aims to identify VOCs linked to oxidative stress employing an in vitro model of alveolar basal epithelial cells exposed to hyperbaric and hyperoxic conditions. In addition, the feasibility of this in vitro model for POT biomarker research was evaluated. The hyperbaric exposure protocol, similar to the U.S. Navy Treatment Table 6, was conducted on human alveolar basal epithelial cells, and the headspace VOCs were analyzed using gas chromatography–mass spectrometry. Three compounds (nonane [*p* = 0.005], octanal [*p* = 0.009], and decane [*p* = 0.018]), of which nonane and decane were also identified in a previous in vivo study with similar hyperbaric exposure, varied significantly between the intervention group which was exposed to 100% oxygen and the control group which was exposed to compressed air. VOC signal intensities were lower in the intervention group, but cellular stress markers (IL8 and LDH) confirmed increased stress and injury in the intervention group. Despite the observed reductions in compound expression, the model holds promise for POT biomarker exploration, emphasizing the need for further investigation into the complex relationship between VOCs and oxidative stress.

## 1. Introduction

Pulmonary oxygen toxicity (POT), an alveobronchial condition that develops after sustained exposure to an elevated partial pressure of oxygen (pO_2_) by the damaging effect of reactive oxygen species (ROS), has been the topic of various hyperoxic and hyperbaric studies over the years [1,2,3,4]. POT can start at a pO_2_ of 51 kPa (0.5 atmosphere absolute [ATA]) and causes symptoms ranging from retrosternal discomfort and burning sensations on inspiration, to acute respiratory distress syndrome and interstitial lung fibrosis if hyperoxic exposure is continued after the first symptoms arise, typically after several hours to days [5]. Patients repeatedly receiving hyperbaric oxygen therapy, or when ventilated in the intensive care unit (ICU) for multiple days with elevated pO_2_, are at risk of commencing POT. Also, recreational divers, for instance, when using rebreathers with hyperoxic gas settings, or commercial or military divers exposed to hyperbaric hyperoxic conditions, can develop POT.

Unfortunately, there is no accurate method used to measure POT. In the late 1960s, the Unit Pulmonary Toxicity Dose (UPTD) was developed, based on a decrease in vital capacity (VC) after hyperbaric hyperoxic exposures in hyperbaric chambers, to quantify hyperoxic exposure [6]. One UPTD equals roughly an exposure to 100% oxygen for 1 min at 101 kPa (1 ATA). The safe daily limit for diving was set at 615 UPTD, corresponding to a 2% decrease in VC in 50% of the population [7]. However, this threshold for POT lies well within the normal 5% day-to-day variation of the VC [8] and was developed after dry hyperbaric chamber exposures, which limits its applicability for divers. 

Previous studies by the Royal Netherlands Navy Diving and Submarine Medical Centre (DMC) have tried to identify biomarkers to detect POT. Various pulmonary test modalities have been investigated, ranging from pulmonary diffusion capacity and nitric oxide measurements to molecular analysis of exhaled breath [4]. Notably, the measurement of volatile organic compounds (VOCs) in exhaled breath seems a promising modality to measure (pre)clinical POT. This non-invasive technique has also been used in pulmonary laboratories to differentiate between various types of asthma or as an experimental diagnostic test for various other pulmonary diseases [9,10,11]. In hyperbaric and diving medicine research, VOCs after various hyperbaric hyperoxic exposures in humans have been identified and compiled into a catalogue known as the VAPOR library [3]. However, most of these exposures took place in the context of regular diving activities and treatment table sessions, staying within pO_2_ limits that did not induce severe POT. Although POT is likely a condition on the spectrum of oxygen exposure that starts subclinically before symptoms are noted, the previously identified VOCs can currently only be associated with hyperoxia rather than actual oxygen toxicity, given the lack of VOC research in patients with severe POT. Ethical constraints, however, restrict the implementation of more extreme exposures than those in previous in vivo studies, thereby limiting the investigation of VOCs during fulminant POT. This led to the development of the current in vitro model. It is worth mentioning, as exhaled breath encompasses VOCs originating from the entire body, that VOCs might be linked to hyperoxic stress in the lungs, but also from other organs. Analyzing VOCs emitted specifically from pulmonary cells could offer a method to differentiate the sources of VOCs, thereby confirming pulmonary involvement. 

The primary objective of this pilot study is to determine whether VOCs linked to oxidative stress can be detected in the headspace of in vitro cell cultures following hyperbaric and hyperoxic exposure in a recompression chamber. The secondary goal is to assess the feasibility of employing this in vitro model for validation and, ideally, as a substitute for in vivo exposures currently constrained by ethical limitations.

## 2. Materials and Methods

### 2.1. Preparation of the Cell Lines

The in vitro model used in this study was established in line with a previous study [2]. In short, human alveolar basal epithelial (A549) cells (CCL-185) were cultured in Roswell Park Memorial Institute (RPMI) 1640 medium (Gibco, ThermoFisher Scientific, Waltham, MA, USA) that was complemented with fetal bovine serum, penicillin-streptomycin, L-glutamine, gentamicin, and amphotericin. A549 cells were grown in an incubator at 37 °C and 5% CO_2_ until they were 85–90% confluent. Prior to each experiment, cells were passed into separate glass headspace vials (Markes International, Cincinnati, OH, USA) in order to limit plastic VOC contaminates by seeding 1.5 × 105 cells in 1 mL of supplemented RPMI-1640 [2,12]. Cells were then incubated for 24 h at 37 °C and 5% CO_2_ until the experimental exposure was initiated.

At the start of the experimental day, the initially seeded RPMI-1640 was obtained for baseline cytotoxicity measurements, described later, and replenished with 200 µL of fresh supplemented RPMI-1640. Subsequently, all vials were sealed airtight as described previously [2] and divided into two groups: an intervention group for hyperbaric exposure to 100% oxygen, and a control group exposed to air (21% oxygen) under the same hyperbaric conditions. The experiment was conducted on three separate days.

### 2.2. Hyperbaric Exposure, Oxygenation, and Sampling

The multiplace recompression chambers (Haux Life Support, Karlsbad, Germany and Pommec-Hytech, Raamsdonkveer, The Netherlands) of the DMC were employed to pressurize the vials. The hyperbaric exposure was based on the U.S. Navy Treatment Table 6 (641 UPTD), with a maximum pressure of 283 kPa (2.8 ATA; equivalent to 18 m of sea water [msw]) and lasted four hours and two minutes, excluding intermittent air periods due to concerns that these periodic flushes with compressed air might eliminate VOCs from the headspace. To compensate for the missing UPTD from the air periods, two minutes were added at 192 kPa (1.9 ATA or 9 msw), resulting in a calculated UPTD count of 642. See Figure 1.

Before hyperbaric exposure, a 0.6 mm diameter hypodermic needle was inserted through the membrane of the cap of all vials to ensure equalization of the vials while pressurized. All vials were placed in an incubator oven (Steinberg Systems, Berlin, Germany) in the hyperbaric chamber and were kept at 37 °C during the hyperbaric exposure and subsequent sampling period.

The vials were pressurized to 283 kPa (2.8 ATA or 18 msw). After arriving at depth, the intervention group vials were flushed for 30 s with 100% oxygen with a 0.6 mm diameter hypodermic needle until the headspace was filled with 100% oxygen, confirmed by an on-site oxygen sampler (Normocap 200, Datex-Ohmeda/GE Healthcare, Chicago, IL, USA). The remaining vials served as control group. After flushing the test vials with 100% oxygen, all needles were removed from the membranes, restoring the airtight seal to prevent an influx of chamber atmosphere and other contaminants into the vials. Just before decompression from 283 kPa to 192 kPa (1.9 ATA; equivalent to a depth of 9 msw), VOC-capturing high-capacity polydimethylsiloxane sorbent fibers (HiSorb^TM^, Markes International Ltd., Bridgend, UK) were inserted through all vial seals. Two hours after completing the hyperbaric exposure, the HiSorb^TM^ probes were removed from the vials. VOCs were sampled up to two hours after completing hyperbaric exposure for a total of five hours and two minutes from the headspace. At the end of the sampling period, the HiSorb^TM^ probes were removed, cleaned, and transferred into empty desorption tubes (Markes International). A graphic representation of the study protocol is shown in Figure 1. After removing the HiSorb^TM^ probes, cell supernatants were collected to analyze cellular stress levels.

### 2.3. Assessment of Cell Stress

Cellular stress was assessed by measuring the concentrations of the inflammatory marker interleukin-8 (IL8) and injury marker lactate dehydrogenase (LDH) in the cell supernatant at the end of the hyperbaric exposure. Additionally, IL8 and LDH were measured within the cell supernatant at baseline, i.e., before transfer to the recompression chamber. IL8 was measured with an enzyme-linked immunosorbent assay as per manufacturer’s instructions (R&D Systems Inc., Bio-Techne, Minneapolis, MN, USA) and LDH was measured as previously described by Zuurbier et al. [13].

### 2.4. Gas Chromatography–Mass Spectrometry (GC–MS) Analysis

GC–MS analysis of the headspace samples was performed within three days of sampling. Extensive details on thermal desorption and GC–MS analysis have been described previously [2]. Similarly, thermal desorption was performed by heating desorption tubes to 250 °C for five minutes while purging with a 30 mL/minute flow in a Markes TD100 autosampler and desorber (Cincinnati, OH, USA). Desorbed VOCs were then captured in a cold trap at 25 °C and subsequently re-injected by rapidly heating the cold trap to 280 °C for one minute. The injection was performed splitless via a transfer line at 180 °C with a flow of 1.2 mL/minute onto an Inertcap 5MS/Sil GC column (30 m, ID 0.25 mm, film thickness 1 μm, 1,4-bis(dimethylsiloxy)phenylene dimethyl polysiloxane [Restek, Breda, The Netherlands]). Temperature of the oven was maintained isothermal at 40 °C for five minutes, subsequently increased to 280 °C at a rate of 10 °C/minute and finally kept isothermal at 280 °C for another five minutes. Ionization of molecules was performed through electron ionization at 70 eV, and fragment ions were detected via a quadrupole MS (GCMS–GP2010, Shimadzu, Den Bosch, The Netherlands) using a scan range of 37 to 300 Da.

### 2.5. Statistical Analysis and Identification of Compounds

Data were compiled and statistically processed using R statistical software (v4.1.2; R Core team 2021), together with additional packages pROC (v1.18.0; Robin et al 2011), epiR (v2.0.61; Stevenson et al 2023), tidyverse (v2.0.0; Wickham et al 2019), MBESS (v4.9.0; Kelley 2022), sva (v3.42.0; Leek and Storey 2012), and rstatix (v0.7.0; Kassambara 2023). All statistical processing was performed with blinded data.

When considering the variations in VOC levels observed in the human TT6 study [14], we anticipated a minimum difference of 10% in intensities between the intervention and control groups. Based on α = 0.05 and a power of 80%, combined with a Mann–Whitney U test, we calculated that a minimum sample size of *n* = 19 per group was required. 

After alignment and preprocessing, multivariate sparse partial least squares discriminant analysis (sPLS-DA) modeling for two components was conducted to identify the most discriminating ion fragments between the intervention and control groups, as described in a prior study [15]. All ions within a retention time of three seconds of each other were considered a cluster belonging to one molecule.

Untargeted identification of the ion clusters using the corresponding peaks in the chromatograms was carried out using GCMS Postrun Analysis software (GC-MS Solution version 4.52, Shimadzu Corporation, Kyoto, Japan), combined with the online National Institute of Standards and Technology (NIST) library [16]. If the peak showed high (>80%) similarity to multiple compounds, further identification was carried out using results of previous studies about breath sample identification and online databases such as the National Center for Biotechnology Information (PubChem) and Human Metabolome Database (HMDB) [3,17,18,19]. If no identical molecules were detected at similar retention time across all different test days, thus only being present on one or two test days out of three, we considered the ion cluster to be a contaminant, resulting in removal of the ions from further statistical analysis. Following the exclusion of all contaminants, the sPLS-DA was reapplied to finalize the selection of ion fragments, and identification as described above was repeated. A Mann–Whitney U test was employed to test the ion clusters for significance between the blinded control and intervention groups, and as a last step, unblinding of the groups was performed. Differences in IL8 and LDH between baseline, intervention group, and control group were tested by Dunn’s test with Holm’s adjustment for multiple comparisons. An α < 0.05 was considered statistically significant.

## 3. Results

During 3 test days, a total of 24 oxygenated vials and 23 control vials were pressurized according to the trial protocol. Due to sampling, collection, or GC–MS analysis errors, 23 oxygenated samples and 21 control samples could be used for statistical analysis. Initial multivariate testing resulted in 17 ion samples for component 1, divided over 10 clusters. After chromatogram analysis, discarding the presumed contaminants, reapplying the sPLS-DA, and univariate testing the clusters for significance between the intervention and control groups, three compounds were identified: nonane (*p* = 0.005), octanal (*p* = 0.009), and decane (*p* = 0.018), see Figure 2, Figure 3 and Figure 4.

After unblinding, all compounds displayed a lower signal intensity in the intervention group compared to the control group. In contrast, the intervention group showed a higher degree of cellular stress and injury at the end of the sampling period, as observed by the significantly higher IL8 and LDH levels compared to the control group, see Figure 5.

## 4. Discussion

### 4.1. Identified Compounds and Signal Intensities

Three compounds (nonane, octanal, and decane) were identified, which varied significantly in signal intensity between the intervention and the control groups. Two of the three compounds were also identified in an in vivo study with similar exposure, affirming the viability of this in vitro model and achieving the primary and secondary objectives of this study. Cell stress markers (IL8 and LDH) demonstrated a clear difference between the intervention and control groups.

As in previous studies with hyperbaric oxygen exposure in humans, predominantly (methyl)alkanes and aldehydes were identified [3,20,21]. Straight-chain alkanes are thought to originate from cell membrane destruction when the polyunsaturated fatty acids (PUFAs) in the membrane are destroyed by pathogens, mechanical stress, inflammatory processes, and chemicals such as reactive oxygen species (ROS), which cause lipid peroxidation [22]. In human studies, decane and nonane have been associated with asthma and various autoimmune diseases [11,23]. Both have also been found in previous hyperbaric hyperoxic studies by our group [14,21]. Most notably, both alkanes were also identified in the in vivo study with a similar hyperoxic exposure [14], indicating the potential usefulness of this in vitro model in the search for VOCs indicative of POT. In contrast, methylcyclohexane, a cycloalkane identified in previous in vivo hyperbaric exposures and considered as an indicator of secondary inflammation following cell damage, was not detected in this study, reinforcing the belief that this compound does not originate from direct cell membrane destruction as is the case with straight-chain alkanes [3,14,21].

Aldehydes are formed during various physiological processes, such as the metabolization of alcohols, but also derive as a secondary product of lipid peroxidation [24,25,26]. Octanal has been identified as a potential biomarker in VOC studies of lung cancer, smoking, and COPD [27,28,29]. However, the in vivo study did not identify octanal with a similar exposure [14].

Nevertheless, the signal strengths of all molecules were higher in the control group than in the intervention group. This observation is consistent with findings from two separate studies that subjected similar lung epithelial cells to hydrogen peroxide (H_2_O_2_) or oxygen concentrations exceeding 90% for a duration of 24 h, inducing substantial oxidative stress [2,30]. In one study, the intervention group showed reduced octane levels compared to the control group, while 2-ethyl-1-hexanol exhibited heightened signal intensity [2]. Although both compounds were detected in our study, no significant differences in signal intensities were noted between the intervention and control groups. The other study also reported a decrease in decane, mirroring our findings [30].

Lower alkane signal intensities after hyperbaric hyperoxia are not limited to in vitro studies. Similar results have been documented by our group in in vivo studies, including a study with subjects exposed to a U.S. Navy Treatment Table 6, identical to the exposure in the in vitro experiment described [14]. Fothergill et al., while employing exhaled breath condensate analysis to detect molecules in breath, also reported a reduction in predominantly alkane levels after a hyperoxic exposure 6.5 h to a pO_2_ of 202 kPa (2 ATA; or 10 msw) in U.S. Navy divers [31]. Conversely, for divers exposed for 7.5 h to a COMEX-30 treatment table with maximum pO_2_ levels of 283 kPa (2.8 ATA; of 18 msw), all VOCs, including several alkanes, demonstrated an increase in expression [32]. The mechanism underlying this counterintuitive decline in alkane expression during limited hyperoxic exposures, as opposed to more extensive exposures with an increase in alkanes, remains unclear. The involvement of redox reactions with free oxygen radical intermediates or further metabolization into other molecules might explain this discrepancy [24,31]. However, no increased levels of end-product metabolites were identified in this study, leaving this explanation unresolved. Alternative explanations could be attributed to a temporary burst in cellular antioxidant capacity and a transient enhancement of membrane protection and stabilization mechanisms, achieved by disrupting other processes affecting the cell membrane [33,34]. During longer exposures, this compensatory mechanism might become depleted, leading to increased cell damage and a subsequent rise in expelled alkanes. Further research is necessary to provide a suitable explanation for this phenomenon of diminishing VOCs with oxidative stress.

### 4.2. Strengths and Limitations

To our knowledge, this study is the first to evaluate the impact of hyperbaric oxygen on in vitro human lung cells through VOC measurement. While A549 cell cultures have been previously subjected to hyperbaric oxygen, it was within the context of preconditioning them before exposure to a mutagenic agent [35]. Another notable aspect is that we conducted all procedures ourselves, from cultivating cell lines to analyzing VOCs, ensuring comprehensive control and understanding of each step and facilitating potential replication of the methodology in subsequent studies. Additionally, this study employs a methodology similar to a previous in vivo study [14] for inducing hyperoxia and evaluating oxygen toxicity, thereby streamlining the comparison of results.

However, several limitations of this study require consideration. This pilot study utilized a monoculture of alveolar basal epithelial (A549) cells. However, the healthy alveolus, where the pathophysiological effects of POT occur, consists of alveolar type I and II cells [36,37]. Being thin, type I cells cover the majority of the alveolar surface, while type II cells, less prevalent in healthy alveoli, are bulkier and mainly responsible for surfactant production. Following alveolar injury, such as sustained POT, type I cells can be replaced by type II cells, leading to thickening of the alveolar-capillary membrane and hindered gas diffusion [38]. In cases of severe alveolar injury, a faulty repair process may result in the growth of bronchial basal epithelium, with subsequent loss of alveolar epithelium and further impairment of gas diffusion [39]. This experiment employed alveolar basal epithelial cells due to the availability of this in vitro model rather than constructing a new model with exclusively alveolar cells. While the phospholipid bilayers of alveolar and bronchial epithelial cell membranes are generally similar, distinctions exist, including the production of surfactant proteins and phenotype [40]. Consequently, the employment of bronchial cells, rather than alveolar cells, may yield different VOCs. For future research, it is advisable to incorporate both types (I and II) of alveolar epithelial cells.

Another limitation associated with in vitro experiments is the reliance on a monoculture of cells, in contrast to the intricate assembly and dynamic interactions found in a normally functioning healthy organ. While the cell medium fluid endeavors to simulate similar conditions, it remains distinct from the support provided by an organ system and its protective natural microenvironment. This is particularly evident in lung tissue, where antioxidant enzymes like superoxide dismutase or catalase, as well as direct antioxidants such as vitamin C, E, and glutathione, are also present in the epithelial lining fluid in the alveoli [41]. This offers enhanced protection against exogenous oxygen radicals. In contrast, in in vitro experiments, where these additional protective environments are absent, hyperoxic damage to the cells could be more pronounced compared to in vivo conditions.

## 5. Conclusions

The model demonstrated an increase in cell stress and injury in the oxygenated intervention group compared to the control group. The three identified compounds (nonane, decane, and octanal) have been previously recognized in studies related to pulmonary diseases and hyperbaric hyperoxic exposures. Consequently, we have successfully demonstrated that this in vitro cell culture model within a hyperbaric chamber is capable of detecting VOCs associated with hyperoxic damage. However, an observed reduction in the expression of these compounds, consistent with findings from earlier studies, and not entirely comprehended, was noted. Given that two out of three VOCs (nonane and decane) were also detected in an in vivo study with similar hyperoxic exposure, we conclude that this model could prove valuable in the search for POT biomarkers, while acknowledging the limitations inherent in a simplified in vitro setting.

## Figures and Tables

**Figure 1 metabolites-14-00281-f001:**
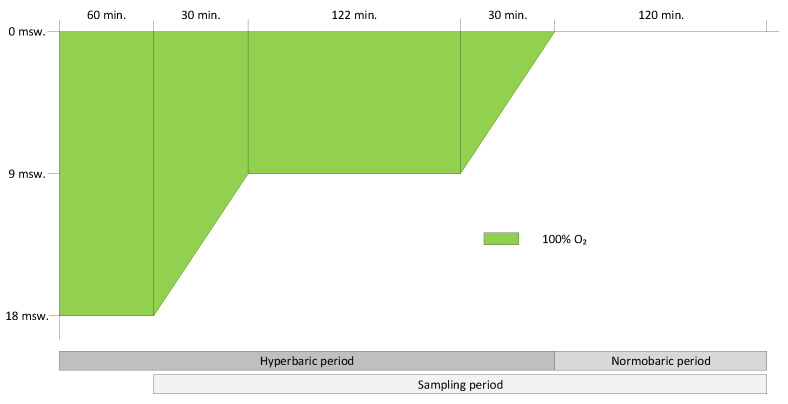
Hyperbaric protocol for the intervention group with 100% oxygen. The control group was exposed to an identical protocol, but with air instead of 100% oxygen.

**Figure 2 metabolites-14-00281-f002:**
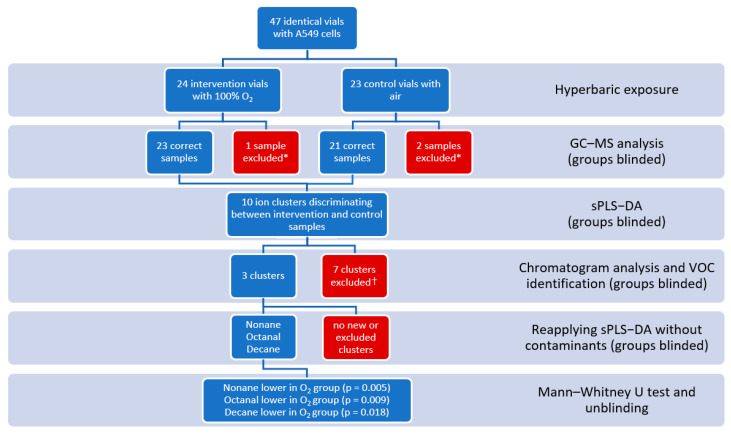
Overview of the methodology, including statistical and identification steps. * Exclusion due to GC–MS or handling errors, resulting in nonfunctional GC–MS output. † Exclusion due to no corresponding peaks in the chomatograms, low (<80%) similarity during identification, or the molecule being identified as a contaminant.

**Figure 3 metabolites-14-00281-f003:**
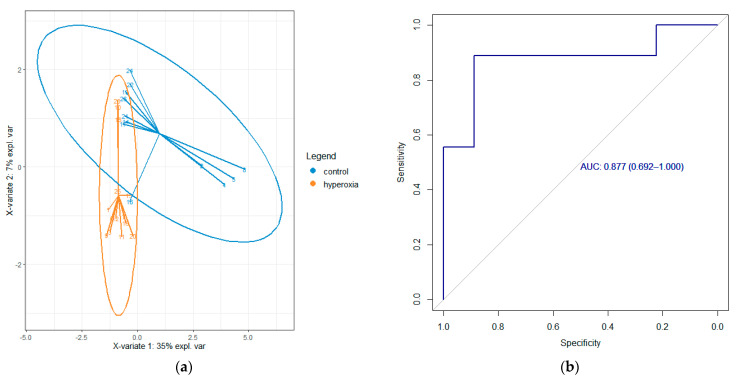
sPLS−DA plot expressing the control and hyperoxic intervention group (**a**), and Receiver Operator Characteristic (ROC) curve and AUC for component 1, consisting of nonane, decane, and octanal (**b**).

**Figure 4 metabolites-14-00281-f004:**
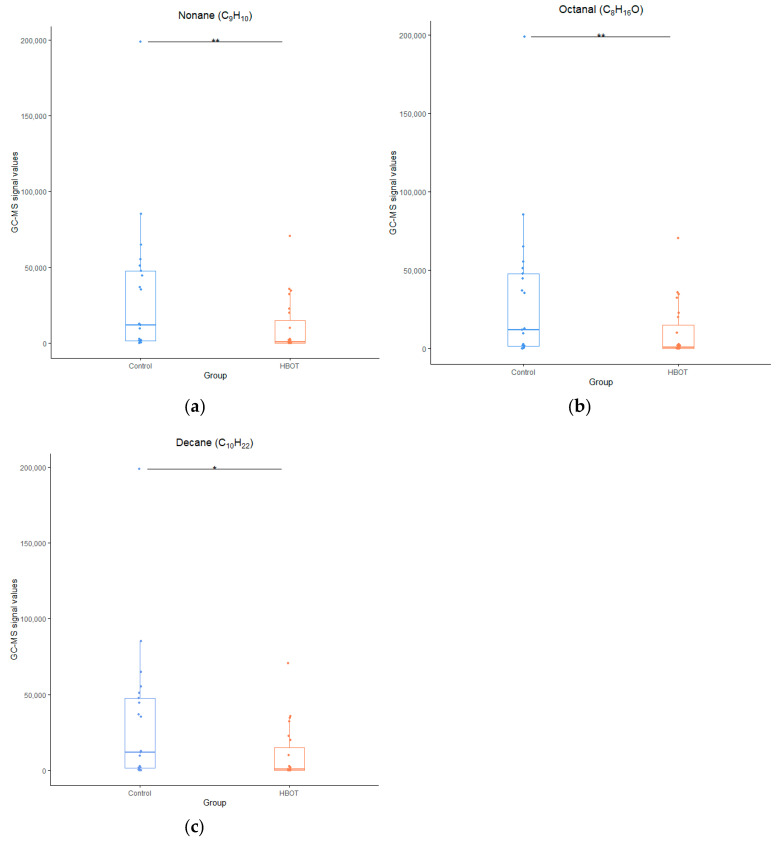
GC–MS signal intensities for nonane (**a**), octanal (**b**), and decane (**c**). Median and interquartile range are represented by the boxplot, with points representing data from individual samples. * *p* < 0.05; ** *p* < 0.01. HBOT: intervention group exposed to 100% oxygen.

**Figure 5 metabolites-14-00281-f005:**
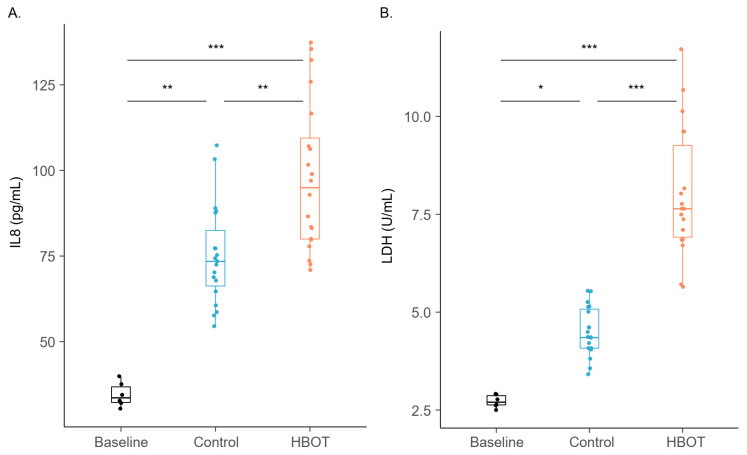
Cell stress and injury levels assessed by IL8 (**A**) and LDH (**B**). Median and interquartile range are represented by the boxplot, with points representing data from individual samples. * *p* < 0.05; ** *p* < 0.01; *** *p* < 0.001. HBOT: intervention group exposed to 100% oxygen.

## Data Availability

The data presented in this study are available upon reasonable request from the corresponding author. The data are not publicly available because the data belong to the Netherlands Ministry of Defence, and therefore cannot be shared unconditionally.

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
