# Peer review of "Volatile Organic Compounds in Cellular Headspace after Hyperbaric Oxygen Exposure: An In Vitro Pilot Study"

_metabolites, 2024, doi:10.3390/metabo14050281_

Round 1

Reviewer 1 Report

Comments and Suggestions for Authors

The manuscript describes usage of volatile organic compounds, emitted from headspace of in vitro alveolar basal epithelial cells following hyperbaric and hyperoxic exposure in a recompression chamber, as biomarkes of pulmonary oxygen toxicity.

The work maintains scientific and approachable manner. The manuscript is technically appropriate, and the data support the conclusions. The authors accurately pointed the limitations of their study. The manuscript requires minor interpunction checking and correction of minor spelling errors, as listed below:

Lines 43 and 51: the same sentence is used to start new paragraphs. Please use different.

Line 46: missing dot to end the sentence: “exposure [10] One UPTD”.

Line 62 and many others: “in vivo” and “in vitro” should be italicized in the whole manuscript

Line 80 and others: between value and degree symbol “°C” should be no space, this is only applied for Kelvins, like “7 K”.

Figure 1: is in the y-axis should be written “msw”, instead of “mtr.”?

Line 167: please provide version number of NIST library.

Line 233: please use lower indices in the formula of hydrogen peroxide.

Based on article evaluation, I recommend minor revision of this manuscript.

Comments on the Quality of English Language

The manuscript requires minor interpunction checking and correction of minor spelling errors.

Author Response

Dear Reviewer,

On behalf of the research team,

Best regards,

Feiko de Jong

Reviewer 2 Report

Comments and Suggestions for Authors

This pilot study provides preliminary insights into in vitro experiments, which, although in some cases seems to confirm what has been observed in vivo, require further confirmation studies. The statistical analysis section should be explained more thoroughly, as well as the experimental setup, possibly through a summary table. Further analyses of biological implications are to be considered after thoroughly revision of the statistical methods (expecially the selection of significant variables)

Major points

·       paragraph 2.2. Hyperbaric exposure, oxygenation and sampling:

o    it coud be useful to add a table to resume the experimental plan including sampliing number, and the differences among baseline, control and HBOT

·        paragraph 2.5. staistical analiysis and identification compounds:

o   This paraghraph shoud be better explained in order to understand data processing  procedures and variable selection criteria

·       Results: it is not clear the validation method

Minor points:

·       introduction - line 43 and 50  t

o   there is the same sentence: “unfortunaly, there is no accurate method to measure POT.

Author Response

(The authors gave the same response as above.)

Reviewer 3 Report

Comments and Suggestions for Authors

Based on the provided manuscript, the ideas provided by the authors are:

- Comparative Analysis of VOCs in Different Experimental Settings: You could delve into a detailed comparison of volatile organic compounds (VOCs) identified in this in vitro study with those found in previous in vivo studies. Analyze the similarities and differences in VOC profiles under different experimental conditions, such as hyperbaric oxygen exposure durations and concentrations, and discuss their implications for understanding oxidative stress and lung injury mechanisms.

- Interpretation of Signal Intensity Discrepancies: Explore the reasons behind discrepancies in signal intensities of identified compounds between the intervention and control groups. Investigate potential factors influencing VOC levels, such as cell stress markers, metabolic processes, and antioxidant mechanisms, and discuss their significance in the context of oxidative stress and cellular damage.

- Evaluation of Methodological Strengths and Limitations: Conduct a thorough evaluation of the strengths and limitations of the experimental methodology employed in this study. Discuss the advantages of conducting all procedures in-house, from cell culture to VOC analysis, and how it enhances control and reproducibility. Additionally, critically assess limitations such as the use of a monoculture of alveolar basal epithelial cells and its implications for mimicking physiological conditions accurately.

- Future Directions and Research Recommendations: Propose potential avenues for future research based on the findings and limitations of this study. Suggest experimental modifications, such as incorporating different cell types or refining experimental protocols, to address existing limitations and deepen understanding of the effects of hyperbaric oxygen on lung cells. Discuss the importance of interdisciplinary approaches and collaboration to advance knowledge in this field.

- Clinical Implications and Translation to Human Health: Discuss the clinical relevance of the study findings and their potential implications for understanding pulmonary oxygen toxicity (POT) and related respiratory conditions in humans. Explore how insights gained from in vitro experiments can inform the development of preventive strategies or therapeutic interventions for mitigating oxidative stress-induced lung injury in clinical settings.

These ideas provide a starting point for further exploration and discussion of the text's content and implications. 

Author Response

(The authors gave the same response as above.)

Round 2

Reviewer 2 Report

Comments and Suggestions for Authors

The authors fullfilled all the reviewer's requests, and the article can be accepted in the present form.